# Correlation between Molecular Docking and the Stabilizing Interaction of HOMO-LUMO: Spirostans in CHK1 and CHK2, an In Silico Cancer Approach

**DOI:** 10.3390/ijms25168588

**Published:** 2024-08-06

**Authors:** Antonio Rosales-López, Guiee N. López-Castillo, Jesús Sandoval-Ramírez, Joel L. Terán, Alan Carrasco-Carballo

**Affiliations:** 1Laboratorio de Elucidación y Síntesis en Química Orgánica, Instituto de Ciencias, BUAP, Puebla 72570, Mexico; antonio.rosalesl@alumno.buap.mx (A.R.-L.); guiee.lopezc@alumno.buap.mx (G.N.L.-C.); jesus.sandoval@correo.buap.mx (J.S.-R.); 2Laboratorio de Modificación y Síntesis en Productos Naturales, FCQ, BUAP, Puebla 72570, Mexico; 3Centro de Química, Instituto de Ciencias, BUAP, Puebla 72570, Mexico; 4CONAHCYT, LESQO, ICUAP, BUAP, Puebla 72570, Mexico

**Keywords:** natural compounds, ADMETx, anticancer criteria selection, molecular dynamics, density functional theory, CHKs

## Abstract

Checkpoint kinases 1 and 2 (CHK1 and CHK2) are enzymes that are involved in the control of DNA damage. At the present time, these enzymes are some of the most important targets in the fight against cancer since their inhibition produces cytotoxic effects in carcinogenic cells. This paper proposes the use of spirostans (Sp), natural compounds, as possible inhibitors of the enzymes CHK1 and CHK2 from an in silico analysis of a database of 155 molecules (S5). Bioinformatics studies of molecular docking were able to discriminate between 13 possible CHK1 inhibitors, 13 CHK2 inhibitors and 1 dual inhibitor for both enzymes. The administration, distribution, metabolism, excretion and toxicity (ADMETx) studies allowed a prediction of the distribution and metabolism of the potential inhibitors in the body, as well as determining the excretion routes and the appropriate administration route. The best inhibition candidates were discriminated by comparing the enzyme-substrate interactions from 2D diagrams and molecular docking. Specific inhibition candidates were obtained, in addition to studying the dual inhibitor candidate and observing their stability in dynamic molecular studies. In addition, Highest Occupied Molecular Orbital—Lowest Unoccupied Molecular Orbital (HOMO-LUMO) interactions were analyzed to study the stability of interactions between the selected enzymes and spirostans resulting in the predominant gaps from HOMOCHKs to LUMOSp (Highest Occupied Molecular Orbital of CHKs—Lowest Unoccupied Molecular Orbital of spirostan). In brief, this study presents the selection inhibitors of CHK1 and CHK2 as a potential treatment for cancer using a combination of molecular docking and dynamics, ADMETx predictons, and HOMO-LUMO calculation for selection.

## 1. Introduction

Checkpoint kinases 1 and 2 (CHK1 and CHK2) have great relevance as pharmacological targets, mainly associated with cancer. Their activity is due to being directly associated with a response of cancer cells to DNA damage; so, selective inhibitors are mainly sought against those kinases [1,2,3]. CHK1 inhibitors (Figure 1) have been directly associated with therapies coupled with DNA damage since CHK1 mediates the DNA repairing process by genes RAD51, FANCE, and the kinase enzyme system DNA-PK, as well as the control of replication by checkpoints Intra-S and G2/M. On the other hand, CHK2 is associated with DNA repair via BRCA1 and cell cycle control via G2/M and G1/S [4,5,6,7,8]. Therefore, the inhibition of CHK1 and CHK2 becomes of interest either as an adjuvant for chemotherapy treatments or in looking for compounds that have a cytotoxic effect on cancer cells, inhibiting their survival pathways.

There are several preclinical inhibitors for CHK1 and CHK2 (Figure 1), such as Isogranulathimide, CCT241533 (which is reported to show dual inhibition), SAR-020106, and PD 407824 [9,10,11]. The last two have a marked preference for CHK1, unlike VRX0466617, which causes both inhibitions. Therefore, this makes its use difficult given its low selectivity between CHKs, as well as its low cytotoxic effect against cancer cells [12,13]. New anticancer compounds are currently being sought, such as CHK1 and CHK2 inhibitors. An example of a cytotoxic steroidal compound with an inhibitory effect on CHK1, previously demonstrated, is diosgenin (Sp1), which by inhibiting this kinase decreases cell viability on osteosarcoma lines via cell cycle arrest [14]. Other with modifications on the nucleus of diosgenin at C6, e.g., bearing substitutions with aryl groups, demonstrating a dual effect. Their glycosylated forms (saponins), decrease the enzymatic activity of CHK1 and CHK2, in addition to inducing apoptosis in cancer cells and a low cytotoxic effect in healthy cells and T lymphocytes. The spirostan nucleus is of interest; it exists in a great chemical diversity (Figure 2), having main modifications in eight positions, ranging from the addition of oxygen atoms to changes in stereochemistry [15,16]. These steroids mostly have an effect against cancer cell lines; however, studies against CHK1 and CHK2 are lacking. For these spirostans (sp) an interesting objective is to search for a specific approach to CHKs, even if the biological evaluation involves a large set of data. The use of bioinformatic tools and Density Functional Theory (DFT) studies allow for analyzing the potential action of a set of molecules for a particular biological activity allowing to the prediction of their pharmacological, pharmacodynamic, and toxicological properties, as well as their stability [17,18].

A set of previously reported synthetic spirostans [19,20] was explored at various levels, firstly, to statistically associate the selectivity between CHK1 and CHK2 in comparison with reference inhibitors. Secondly, the structures with better coupling energy than the reference inhibitors were analyzed, as well as the binding site and the interaction in each case, to explain this possible improvement in inhibition. The DFT results were then analyzed to establish a relationship between stability and interaction sites, to propose stabilizing HOMO-LUMO interactions for a ligand and protein, and to determine their ADMETx properties. From those data, a set of specific inhibitors for CHK1 and CHK2 candidates could be afforded.

## 2. Results and Discussion

### 2.1. Reference Docking

The inhibition of CHK1 and CHK2 is an excellent given alternative against the development of pathologies such as cancer, which interrupts the cell cycle and lowers the viability of cancer cell proliferation, with a low cytotoxic effect [21,22]. In the first phase of this study, known inhibitors for both CHKs were modeled, and particularly for CHK1. The molecule co-crystallized with the protein was also modeled at the site of allosteric inhibition, as seen in Table 1. For CHK1 at the catalytic site, the docking score varies from −3.993 to −6.041 for CCT241533 and Isogranulathimide, respectively. For CHK1, equal or greater coupling in the catalytic site implies a high potential as an inhibitor for this. The same happened at the catalytic site of CHK2, showing that the best inhibitor is VRX0466617, and the one with a lower score was CCT241533, demonstrating the duality of the latter as an inhibitor; for these, it was used as a framework for the selection of new inhibitors in CHK1 and CHK2.

The allosteric site reported for CHK1 is interesting; however, the prediction of a new allosteric inhibitor is complicated, given that a conformational change is predicted in the inactivation of the enzyme. So, studying the interactions of the molecule co-crystallized with the protein bound to the site of allosteric inhibition serves as a framework to predict new candidates. The interactional and spatial analysis allows us to observe the relevance of the active site as well as the strength of the interactions. The controls used in Table 1 are provided in the Appendix A (Appendix A).

### 2.2. Spirostans Docking in CHKs Sites

Table 2 presents the statistical analysis for the docking scores obtained from the database with CHK1 (cat) (Checkpoint kinases 1 catalytic site), CHK2 (cat) (Checkpoint kinases 2 catalytic site), and CHK1 (Allo) (Checkpoint kinases 1 allosteric site), respectively. In Table 2 R(C-#) represents the carbons where the analyzed modifications were presented (Figure 3). Upon obtaining a non-normal distribution, they were analyzed using a non-parametric Kruskall Wallis test, to determine the significance of the modification in each analyzed R.

For the docking carried out at the catalytic site of CHK1, only C-3 and C-12 present a significant difference in the type of substituent. The first provides a better docking score in the presence of an equatorial hydroxyl or a ketone group. The better coupling is due to the high stability of these groups compared to the presence of a hydroxyl in an axial position. In C-12, the presence of hydrogens instead of ketone eliminates derivatives of the hecogenin family as candidates for catalytic inhibitors of CHK1. For the allosteric site of CHK1, there are several descriptors with statistical relevance, mainly with the presence of hydroxyls in the C-3, C-5, C-6, and C-11 positions. Finally, at the catalytic site of CHK2, functional groups at the positions in the A ring are more relevant, as well as the stereochemistry at C-25. Which improves the coupling of derivatives of the axial methyl group (25*S*). Highlighting these descriptors (position and functional group) as key for future structural optimizations. CCT241533 was a reference inhibitor for CHK1 and PD407824 for CHK2. These were the selection criteria for new inhibitors, particularly from the spirostans database, derived from those reported in nature and their possible permutations (Figure 3), resulting in 13 candidates for inhibition of CHK1 and 13 for the CHK2, with the capacity to inhibit both proteins at their catalytic site (Figure 4). However, none of the spirostans showed a possible effect as an allosteric inhibitor of CHK1.

Analyzing the set of selected structures for each CHK1, a trend can be noted for CHK1 candidates with 3*β*-OH and 3-oxo groups, and hydrogens at C-12 except for Sp97; this can be explained due to the interactions that each one shows with the catalytic site of CHK1 (Figure 5). It is observed that the potential inhibitors are interacting at the catalytic site of the CHK1 enzyme. The 2D diagrams of the interactions of CHK1 with its potential inhibitors (Figure 5b–h) show that the candidates present regions that remain constant among themselves: a hydrophobic nature and a negative charge. The first is because it is, generated by the aliphatic steroid skeleton, and the second is because of the presence of hydroxyl groups, carbonyls, and/or double bonds, placed in different positions. The specific generated interactions are those of hydrogen bonds, produced by the group’s hydroxyl in C-2, C-3, and C-6 and the amino acid residues of Asp148 and Tyr 86. However, in Figure 5b there is a difference between Sp1 and Sp97, and Sp154 and Sp137, where the first fit better to the catalytic site, since more amino acid residues interact with them. The first fit better to the catalytic site, but Sp154 and Sp137 interact only partially with the catalytic site. The substituent at C-2 and C-3 may be a factor that influences this interaction; the possible inhibitors that best fit the catalytic site 3*β*-OH in addition to the effect of 2α-OH. Furthermore, if attention is focused on other interactions, those of positively charged amino acids stand out, such as Arg95 and Lys38, present in all potential inhibitors, as well as polar amino acid residues such as Gln13, Thr14, Gln24, Ser88, Asn135 and Ser147. When comparing all these interactions with those shown by the reference drug CCT241533 (Figure 5b), it can be said that the type of interaction is conserved, due to the presence of positively charged and hydrophobic regions, the polar residues Gln13, Thr14, Ser88, Asn135, Ser147, and the positively charged Lys38. All these amino acid residues are of interest since they will also be present in the reference drug CCT241533, this fact locates the key amino acids at the catalytic site (see Appendix A). With the analysis of the interactions and coupling energies, the discrimination of the proposed inhibitors can be seen. Firstly, it is necessary to analyze interactions at Figure 5h, since Sp137 is a dual inhibitor (CHK1 and CHK2). Regarding specific inhibitors, the candidates could be Sp1 (diosgenin, which agrees with reported datum), Sp3, Sp24, and Sp97.

In Figure 6a, the 3D interaction of the potential inhibitors with CHK2 can be detected, highlighting that all inhibitors reach the enzymatic catalytic site. The 2D diagrams in Figure 6 show the interactions of the amino acid residues of the catalytic site with the possible inhibitors. The hydrophobic region is latent in all the inhibitors, in a similar way that happens with CHK1; the polar, negatively charged, and positively charged regions can be observed. Hydrogen bonds are generated due to the presence of some oxygen atoms of the spiroacetal or substituted hydroxyl groups in different positions and amino acid residues of the catalytic site. For example, in Figure 6c between 3αOH and Asp311; Figure 6d between 3αOH with Glu308, and the oxygen of the E ring with Lys245; in Figure 6e 3αOH with Asp 311; in Figure 6f 2αOH and 3αOH with Asp 311; in Figure 6g 3βOH with Lys245, and 6αOH with Met304; and Figure 6h 6αOH with Glu308. Unlike what was shown for the possible inhibitors for CHK1, for CHK2 none of the candidates fit completely within the catalytic site. In this case, the type of replacement in C-2 and C-3 is not relevant. Since it is placed outside the site, thus preventing the interaction by kinase. Finally, the amino acid residues around the spirostans are conserved (see Figure 6). That is the case for positively charged Lys224, negatively charged Asp311, Glu305, Glu308, and polar Thr225, Ser228, Gln358. These interactions are also present in the reference drug (PD407824), so it is important to point them out, thus finding the key amino acids in the catalytic site (see Appendix A). The first two are expected to have a specific inhibition and the third a dual inhibition for CHK1 and CHK2.

### 2.3. ADMETx Studies

The ADMETx properties of the candidates for competitive inhibitors for CHK1 and CHK2 were analyzed, and the most relevant properties are summarized in Table 3. Lipinski’s criteria show that the selected steroids present one or zero violation, which makes them acceptable for oral administration, and which correlates with the PHOA values. The values for QPlog Po/w are inside the recommended range of −2.0 to 6.5 with a minimum value of 4.574 and maximum of 6.219. These values are close to the upper limit of hydrophobic compounds, this being crucial for their passage through the cell membrane but making it difficult for them to stay in the cytoplasm and to reach the enzymes of interest.

Since the approach for these molecules is directed towards cancer, a low or no effect on the CNS is expected; predicted values from −2 to 2 are acceptable; the spirostans presented values of 0, except for Sp131, which showed −1, which means inactivity. For Sp3, Sp1, and Sp5, a value of 1 means little activity in the central nervous system, and for Sp5, a value of 2 is still acceptable. These values estimate low effects on the CNS by the selected steroids, and continuing in the context of metabolism, the number of possible metabolic reactions for the different spirostans is between 1 and 5, values that are within the recommended range of 1 to 8; those small numbers suggest that the reactions can be more specific. Finally, in terms of metabolism and adverse effects, the QPlogHERG value, that predicts the IC_50_ for potassium channel blockade, maintains values from −4.458 to −3.765, which is important because this value must remain higher than −5, given that higher doses are required to present adverse effects due to the potassium channel block, it is estimated that if lower doses of 10^−5^ are required, the entire potassium/sodium exchange system in the body is affected. On the excretion side, regarding the criterion of skin permeability, spirostans have a minimum QPlogKp value of −3.215 and a maximum of −2.188, which underlies a low bioaccumulation.

### 2.4. HOMO vs. LUMO Studies

It is well known that attractive interactions are all important for any rational approach to drug design. The key amino acids of catalytic site present intermolecular interactions with potential inhibitors; however, given the position, the formation of non-bonding HOMO-LUMO stabilizing interactions from the protein to the ligand and vice versa were found. The assessment of the HOMO and LUMO orbital interactions at the catalytic site and the spirostans, for each CHK, showed a trend that favors the interactions of the HOMO of the protein to the LUMO of the Sp (Figure 7), except at the Glu and Asp residues in both cases, denoting that in both enzymes its site is highly conserved. Particularly for CHK1, only in Gly89, there was a 100% frequency where the same gap predominates; in the other residues of the site, at least one molecule inverted the direction of the gap, particularly for the dual inhibitor Sp137 (Table 4). In contrast, for CHK2, in residues Leu226, Gly227, Lys245, Ala247, Glu306, Gly307, and Leu354, the gap from HOMO_CHK2_ to LUMO_Sp_ predominated in most of the residues of the site, with only a Sp of the gap (Sp137) in the rest, except for Met304, which was two Sp. This indicates that for a dual inhibition the gap of the Sp towards the protein must predominate, while for selective inhibitors (Table 4) the gap direction is inverted.

Figure 8a,b show HOMO-Sp/LUMO-protein and HOMO-protein/LUMO-Sp interactions at the catalytic site. These representations allow to detect the best energetic gaps. For the dual inhibitor, the pair HOMO-Sp/LUMO-protein generate the greatest number of interactions that involve the steroidal A ring, the unsaturated B ring, and the hydroxyl group at C-6. In contrast, in selective inhibitors (Appendix A) there are fewer cases where the gap from the HOMO of Sp to the LUMO of CHK predominates, although these interactions are not limited to rings A and B, but also occur in rings E and F, which is explained given the reactive sites of the Sp. Regarding the inverse direction of the HOMO of the protein to the LUMO of the Sp, the formation of interactions with the entire structure, as well as intermolecular interactions on the part of the protein, indicates that it increases the strength of the coupling not only by intermolecular interactions but also by non-binding stabilizing attractions on the part of the HOMO-LUMO gap, favoring the enzyme-inhibitor interaction.

### 2.5. Molecular Dynamics Studies and Post-MM-GBSA Analysis

The stability of the enzyme-inhibitor complexes is evaluated by molecular dynamics simulation at 100 ns (Figure 9). The fluctuations of the complete enzyme inhibitor and the enzyme are shown in terms of RMSD (Root Mean Square Deviation) protein–ligand and protein, as well as the rGyr (radius of gyration) and the RMSF (Root Mean Square Fluctuation), to show the effect of the formation of the enzyme-inhibitor complex on the enzyme. For the RMSD between the enzyme and the inhibitor, in both reference (CCT241533) and Sp137 and Sp1, the best CHK1 inhibitors, high stability is observed, given that an RMSD of less than 1 Å is observed at 100 ns. In particular, the two steroids maintain the same level of behavior, while CCT241533 is slightly higher in value, denoting that a better docking score indicates better stability, a similar phenomenon that is observed in the global RMSD of the protein, which has values around 2 Å. It takes 8 ns to reach stability at this value, given the freedom of movement of the characteristic loops of the enzyme. The rGyr is relevant to analyze the possible inhibition as a function of time, since it indicates the conformational effect of the protein, with a value around 4.80 for Sp137, 2.00 for Sp1, and 5.85 for CCT241533, indicating that a higher score reflects a greater strength on this parameter. Finally, in terms of the effect on the protein residue (RMSF), it presents an effect in the region of the catalytic site and the terminals of the enzyme.

In terms of contacts with the key amino acids that were discussed above (Figure 10), we can denote that they behave differently between Sp1 and Sp137, as well as the reference inhibitor (CCT241533), conceiving high interactions with Asp148, by the spirostans with Tyr86. The increase in Sp137 is associated with Gly17, Glu85, and Cys87. In Appendix A can be observed the contacts and their stability, as well as the RMSF per ligand, SASA, PSA, torsions, and contacts as a function of time. In addition to this, the coupling energy was determined through an MM-GBSA approach, finding for CCT241533 a ΔG_Binding_*=* −48.153 kcal/mol (−47.9 to −49.6), while for Sp137, ΔG_Binding_*=* −58.378 kcal/mol (−57.2 to −58.9), and for, Sp1 ΔG_Binding_*=* −53.863 kcal/mol (−53.2 to −54.3), as an average with respect to the stability of the complex, denoting the same relationship as that obtained in the docking score with respect to the strength of CHK1 inhibition.

A stable enzyme-inhibitor complex exists, differently than on CHK2 (Figure 11); however, the protein produces many fluctuations on the RMSD, generating an inhibitory effect. We can observe that Sp137 and Sp13 are above and below, respectively, the reference inhibitor (PD407824); in terms of stability, it takes 10 ns to reach the same level for PD407824 and Sp13. Sp137 at 5 ns stabilizes at 5.2 Å for the RMSD, but not until 60 ns does it finally stabilize at 8.1 Å. For rGyr, unlike that observed in CHK1, there is a greater effect on the part of the Sp, with average values of 4.85 Å and 4.95 Å for Sp137 and Sp13, respectively, while for PD407824, the average value is 4.31 Å. Finally, the RMSF, it is at the center of the protein where the greatest effect is generated, and this is a result of the coupling on the outside of the protein, generating its collapse. In terms of inhibition, the high modification of the protein represents an inhibitory effect on it, given that the enzyme–inhibitor complex is stable; however, the protein is affected by said coupling.

Finally, the contact analysis with the key amino acids (Figure 12) denotes a highly polar behavior on the part of the active site and therefore the highest coupling energy of molecules, conserving Glu308 with high interactions on Sp137, Sp13, and CCT241533. Particularly, per molecule, high contacts were presented for each one, being Leu226, Gly227 Met304, and Asp311 for the CCT241533, while for Sp13 it was Asp311, Gly351, and Thr367, showing a more polar trend which increased the coupling energy. However, on the part of Sp137, the same amino acids were highlighted as in the CCT241533 except for Asp311, which indicates a similar behavior only with greater strength due to the bridges with water that are preferentially formed. Studies of RMSF as a function of the SASA ligand, PSA, torsion, and time function contacts can be seen in the supplementary material (Appendix A). Finally, the MM-GBSA calculations found for PD407824 a ΔG_Binding_*=* −51.871 kcal/mol (−51.7 to −52.3), while for Sp137, ΔG_Binding_
*=* −59.912 kcal/mol (−59.6 to −60.1) and for Sp13, ΔG_Binding_
*=* −54.287 kcal/mol (−53.9 to −54.7). The values were slightly higher in the CCT241533 and in Sp137 than those observed in CHK1; this, combined with the larger contact area found for the couplings with CHK2 vs. CHK1, shows the same trend as in the docking score.

## 3. Materials and Methods

### 3.1. Protein Preparation and Validation

Crystals for the CHK1 catalytic site (6FCK, [23]), CHK1 allosteric site (3JVS, [24]), and CHK2 catalytic site (4A9U, [25]) obtained from the Protein Data Bank were prepared in the Protein Preparation Wizard module [26] at physiological conditions of pH 7.4 and minimized with a cutoff of a Root Mean Square Deviation (RMSD) ≤ 0.3 Å with an OPLS4 force field according to the previously reported protocol [27] for the validation of each protein.

### 3.2. Spirostan Preparation

The previously constructed database of 155 spirostans (see Appendix A) [20] was minimized in structure using Macromodel [28] and brought to physiological conditions of pH 7.4 in LigPrep [29] according to the previously reported protocol [27] with an OPLS4 force field.

### 3.3. Density Functional Theory Studies

Density Functional Theory (DFT) studies were performed with B3LYP-D3 at 6-31G** (d,p) basis set theory level using Schrödinger’s Jaguar [30] according a previous report for steroidal derivates [31], and other organic compounds [32,33,34], using an Automatic mode, include Exited state type singlet, number of excited states 1, maximum TDDFT interaction 32, energy convergence threshold 5 × 10^−5^, residual convergence threshold 0.01, Medium grid density, Nonrelativistic Hamiltonian. The polarizable continuum model mas employed for the calculation in solution phase, by PCM in solvent (Fist Methanol and recalculated Water), PCM model: CPCM, PCM raddi: Bondi, to obtaining the Highest Occupied Molecular Orbital (HOMO), and the Lowest Unoccupied Molecular Orbital (LUMO), with a Box size adjustment 0.0 A/side, Grid density 5.0 pts/A for Sp.

### 3.4. Molecular Docking

Molecular docking studies were performed in the Glide module [35] with the flexible ligand technique and flexible Ser, Try, Cys, and Met residues, as well as residues at the interaction site according to the previously reported protocol [27] with an OPLS4 force field.

### 3.5. ADMETx Studies

With the previously optimized structures, their ADME properties were calculated in the Qikprop module [36], and the mean lethal doses were calculated in Gussar [37].

### 3.6. Protein DFT Interaction Site

The protein residues selected in the docking studies were determinate HOMO and LUMO for each. DFT was calculated with B3LYP-D3 at 6-31G** (d,p) basis set theory level using Schrödinger’s Jaguar [30] according a previous report for protein studies [38,39], using an Automatic mode, include Exited state type singlet, number of excited states 1, maximum TDDFT interaction 32, energy convergence threshold 5 × 10^−5^, residual convergence threshold 0.01, Medium grid density, Nonrelativistic Hamiltonian. The polarizable continuum model was employed for the calculation in solution phase, by PCM in solvent Water, PCM model: CPCM, PCM raddi: Bondi, to obtaining the Highest Occupied Molecular Orbital (HOMO), and the Lowest Unoccupied Molecular Orbital (LUMO), with a Box size adjustment 0.0 A/side, Grid density 5.0 pts/A for Sp. The residues that presented interactions with each ligand were prepared for the site defined as interactions for CHK1 and CHK2 for the sectioned calculation of DFT of the protein fragment (amino acid residues) and Sp selected for calculation of HOMO and LUMO at said site in Schrödinger’s Jaguar [40].

### 3.7. Molecular Dynamics

Molecular dynamics (MD) simulation was calculated using the Desmond module by Schrödinger for the selected Sp and reference (CCT241533 and PD407824, respectively) for CHK1 and CHK2 at the catalytic site. It is based on the best result obtained in the molecular docking for each of the enzyme-molecule complexes. The TIP3P water model was used in a solvation system of orthorhombic periodic boundary conditions with a buffer distance of a,b,c = 10 Å. The system Sp or reference (CCT241533 and PD407824) to CHKs complex was neutralized with Na^+^/Cl^−^ ions (0.15 M), and the simulation was run under an isothermal-isobaric ensemble (NPT) at a temperature of 300K and bar pressure of 1.013 with a total of 1000 frames, recording interval (ps) of 100.0, and energy of 1.2. The system minimized their energies by using the Nose–Hoover thermostatic algorithm to maintain the temperature and the Martina–Tobias–Klein method to maintain pressure. The efficiency of the SS binding complex was analyzed by Root Mean Square Deviation (RMSD), Root Mean Square Fluctuation (RMSF) plots, ligand–protein interactions histograms, and MM-GBSA. MD simulations were acquired three times to analyze reproducibility, and file Desmond output performing simulation was analyzed using a simulation interaction diagram for Schrödinger software [41,42].

### 3.8. Binding Free Energy Calculation

The binding energy of the selected Sp and references (CCT241533 and PD407824) were analyzed using Molecular Mechanics with MM-GBSA of prime Schrödinger molecule [41,42]. The solvation model was set to VSGB (Variable Dielectric Surface Generalized Born) and the force field to OPLS4. Protein residues were delimited 12 Å from the ligand for energy minimization. The binding free energy was calculated by:ΔG_Binding_ = G_Complex_ − (G_Protein_ − G_ligand_)(1)
where ΔG_Binding_ = binding free energy, G_Complex_ = free energy of the complex, G_Protein_ = free energy of the target protein and G_ligand_= free energy of the ligand.

The value presented was the result of the average of the 100 ns obtained by MD simulation, as well as its minimum and maximum value intervals [41,43,44].

## 4. Conclusions

Specific inhibitor candidates for CHK1 and CHK2 were determined with the assistance of bioinformatics tools and studies such as molecular docking and predictions of ADMETx properties. For CHK1, possible candidates were spirostans Sp1, Sp3, Sp97; for CHK2, they were Sp13 and Sp28. There is a trend in the presence of non-polar interactions between the catalytic site of both enzymes and possible inhibitors, due to the hydrophobic nature of spirostans. All the studied spirostans fit inside the enzyme pocket to carry out the inhibitory effect due to interaction with the catalytic site of the enzymes. The complete structure of the spirostans does not totally insert into the catalytic site of the enzymes; however, this does not mean there is a problem for them to generate a possible inhibitory effect, as observed in the 2D diagrams. There is a relation between molecular docking and molecular dynamics studies, where a low docking score can be translated into greater stability of the protein-spirostan complex. In particular, for CHK1, the dual inhibitor candidate (Sp1), the best specific candidate for CHK1 (Sp137), and the reference (CCT241533) have high stability when the enzyme-ligand complex is generated; on the other hand, in studies concerning CHK2, the dual inhibitor candidate (S137), the best specific candidate for CHK2 (Sp13), and the reference (PD407824), the stability of the enzyme-ligand complex exists; however, with Sp13 it is less stable over time. In the presence of acidic amino acid residues, the greatest stability by non-binding interactions is given by the gap between HOMO_Sp_ and LUMO_CHK_, but in any other residue the gap is inverted to HOMO_CHK_-LUMO_Sp_.

## Figures and Tables

**Figure 1 ijms-25-08588-f001:**
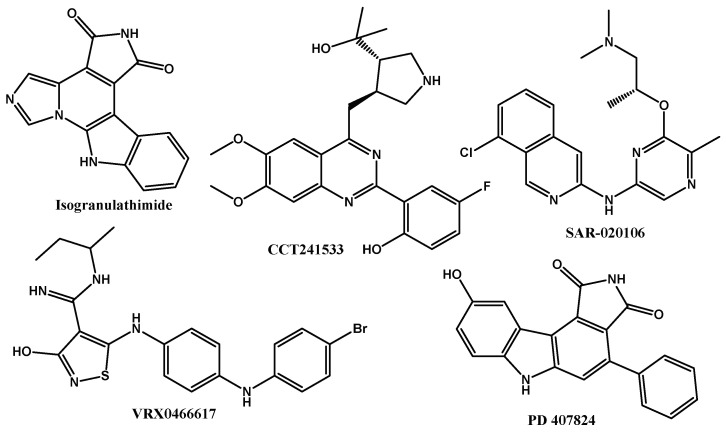
Preclinical inhibitors for CHK1 and CHK2.

**Figure 2 ijms-25-08588-f002:**
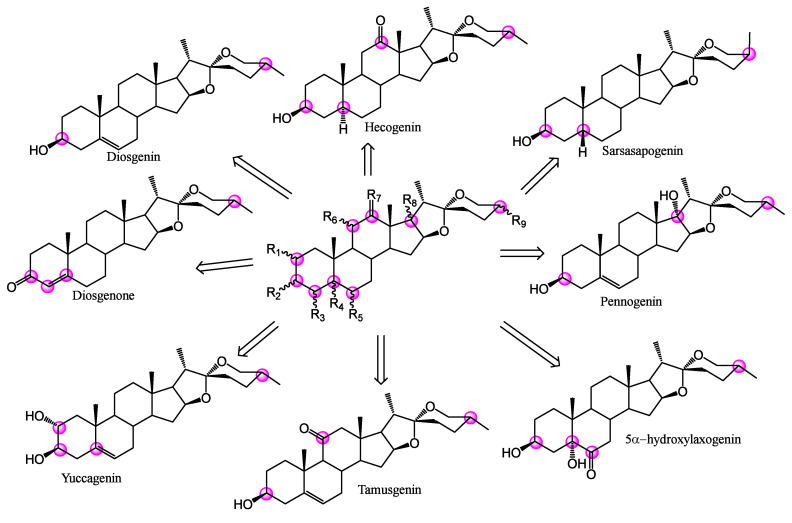
Biological active spirostans.

**Figure 3 ijms-25-08588-f003:**
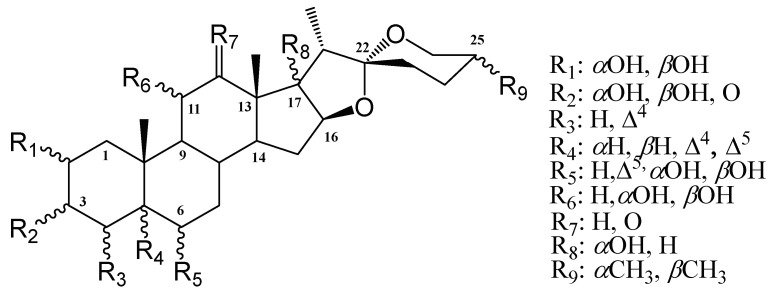
Skeleton and substituent numeration for spirostans and modifications studies.

**Figure 4 ijms-25-08588-f004:**
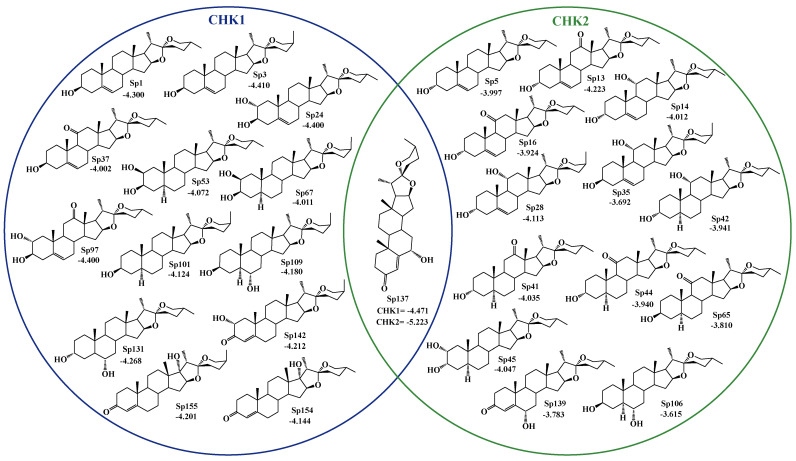
Docking score represented in a Venn diagram for spirostan. Better than CCT241533 for CHK1 and PD407824 for CHK2.

**Figure 5 ijms-25-08588-f005:**
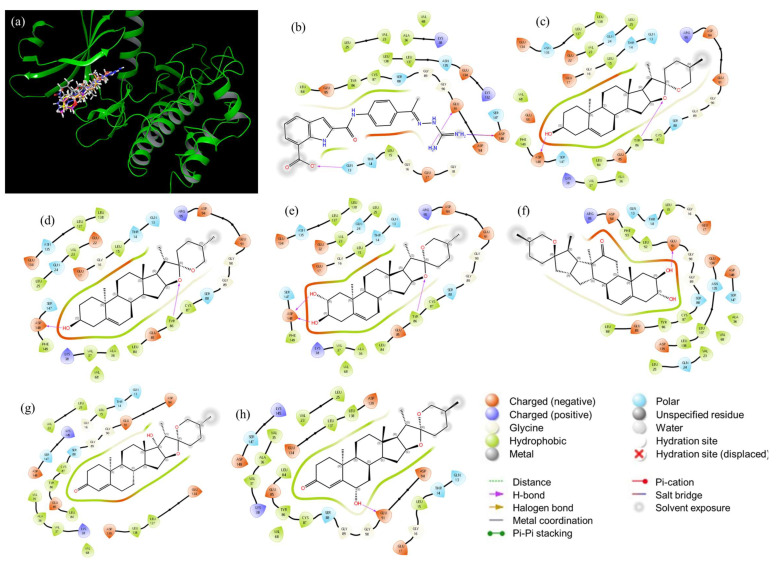
Interaction diagram for CHK1 (catalytic site) for: (**a**) 3d Superposition; (**b**) CCT241533; (**c**) Sp1 (diosgenin); (**d**) Sp3; (**e**) Sp24; (**f**) Sp97; (**g**) Sp154; and (**h**) Sp137.

**Figure 6 ijms-25-08588-f006:**
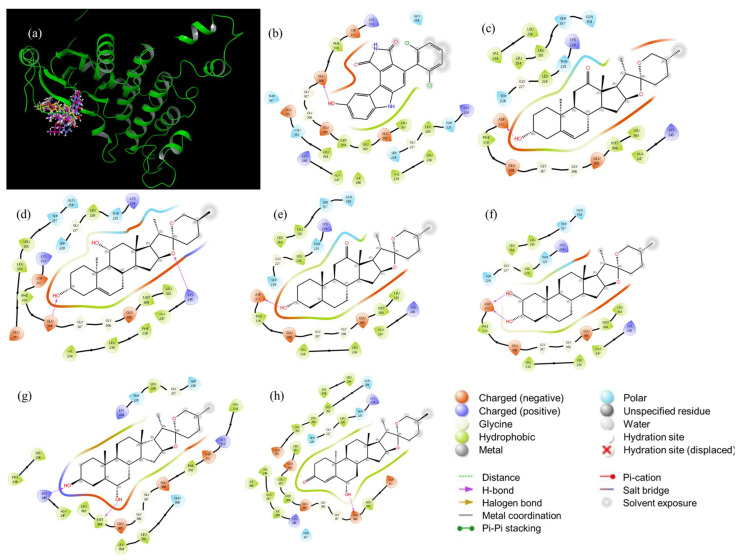
Interaction diagram for CHK2 (catalytic site) for (**a**) 3D superposition; (**b**) PD407824; (**c**) Sp13; (**d**) Sp28; (**e**) Sp41; (**f**) Sp45; (**g**) Sp106; (**h**) Sp137.

**Figure 7 ijms-25-08588-f007:**
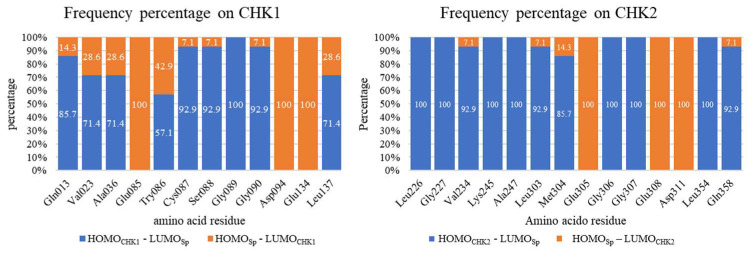
HOMO-LUMO gap frequency percentage in key amino acids of CHK1 and CHK2 vs. spirostans.

**Figure 8 ijms-25-08588-f008:**
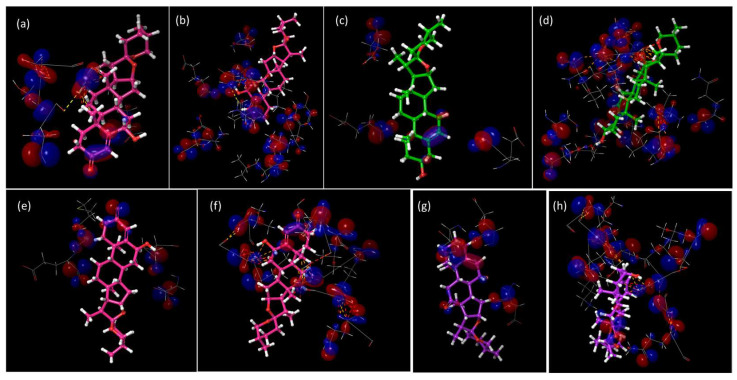
HOMO-LUMO analysis in catalytic site: (**a**) HOMO_Sp137_-LUMO_CHK1_; (**b**) HOMO_CHK1_-LUMO_Sp137_; (**c**) HOMO_Sp1_-LUMO_CHK1_; (**d**) HOMO_CHK1_-LUMOS_p1_; (**e**) HOMO_Sp137_-LUMO_CHK2_; (**f**) HOMO_CHK2_-LUMO_Sp137_; (**g**) HOMO_Sp13_-LUMOC_HK2_; and (**h**) HOMO_CHK2_-LUMO_Sp13_.

**Figure 9 ijms-25-08588-f009:**
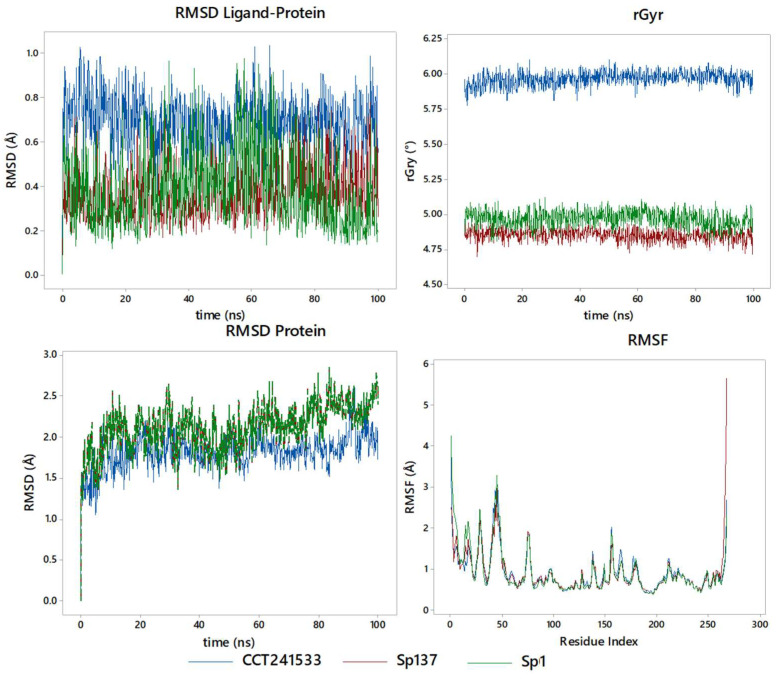
Molecular dynamics studies for CCT241533, Sp137, and Sp1 in CHK1: RMSD of ligand–protein, RMSD protein, and rGry.

**Figure 10 ijms-25-08588-f010:**
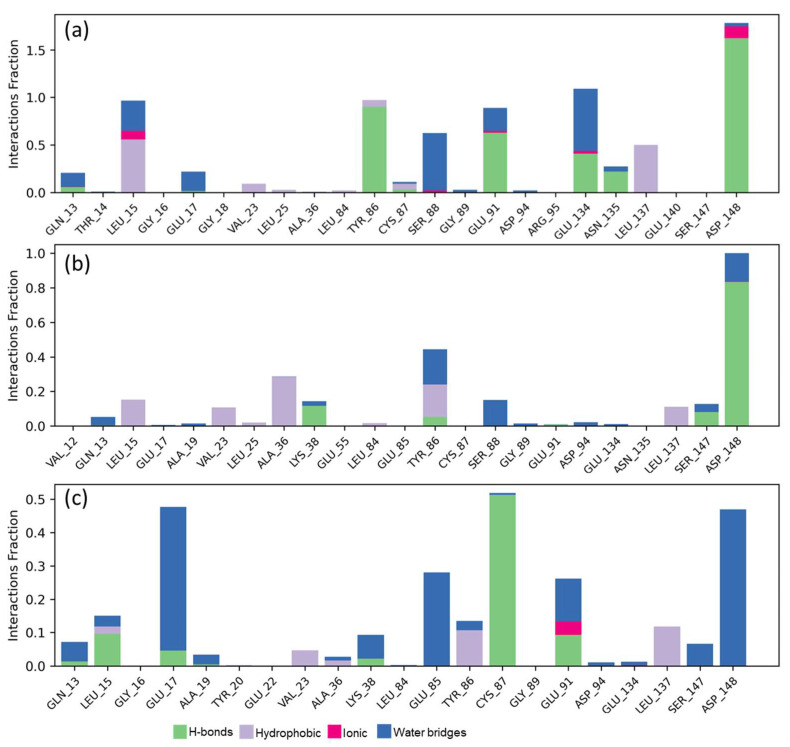
Ligand-protein contacts with CHK1 for (**a**) CCT241533; (**b**) Sp1; (**c**) Sp137.

**Figure 11 ijms-25-08588-f011:**
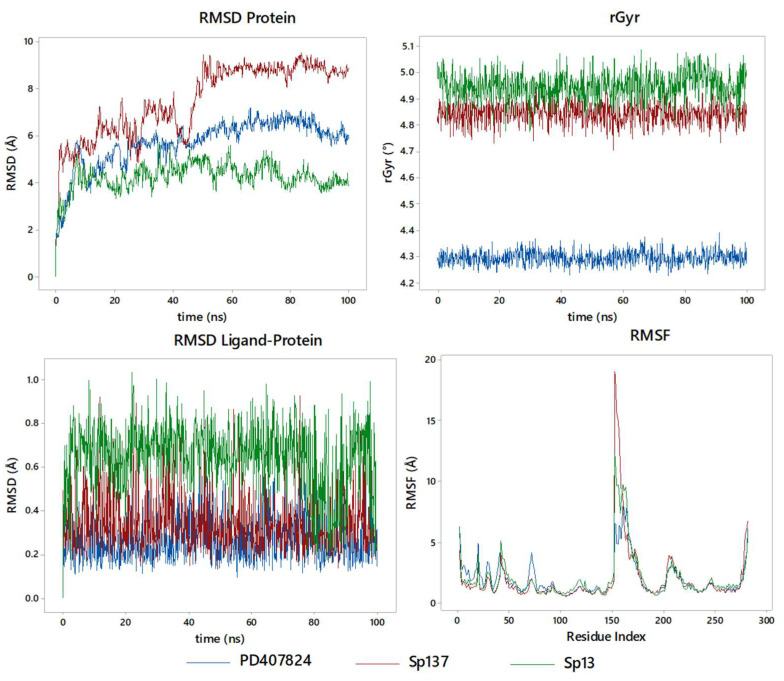
Molecular dynamics studies for PD407824, Sp137, and Sp13 in CHK1: RMSD of ligand–protein, RMSD protein, and rGry.

**Figure 12 ijms-25-08588-f012:**
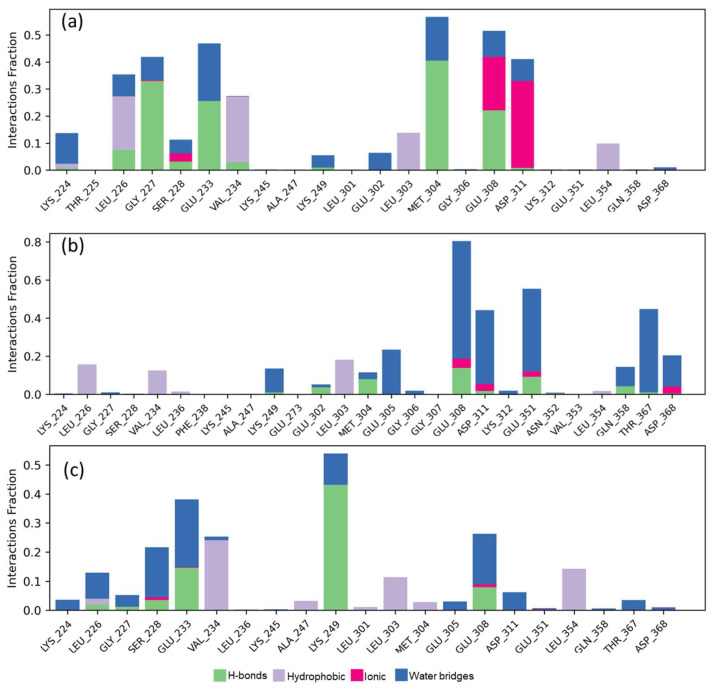
Ligand-protein contacts with CHK2 for (**a**) PD407824; (**b**) Sp13; and (**c**) Sp137.

**Table 1 ijms-25-08588-t001:** Docking score (DS) of reference inhibitors in CHKs.

Preclinical Inhibitor	DS with CHK1		DS with CHK2
	Catalytic	Allosteric	Catalytic
Isogranulathimide	−6.041	-	−5.684
CCT241533	−3.993	-	−5.145
Sar-020106	−6.041	-	−6.021
PD 407824	−4.754	-	−3.261
VRX0466617	−5.900	-	−6.592
Molecule co-crystallized with the protein	-	−6.342	-

**Table 2 ijms-25-08588-t002:** Spirostans in CHKs docking score (DS). Statistical analyses by Kruskall Wallis test.

R(C-#)	CHK1 (Cat) Better DS →		CHK1(Allo) Better DS →			CHK2 (Cat) Better DS →	
1(C-2)	NSD		αOH, *β*OH	H		NSD	
2(C-3)	αOH	*β*OH, O	αOH	*β*OH, O		αOH	
3(C-4)	NSD		H	Δ^4^		H	Δ^4^
4(C-5)	NSD		Δ4, αH, Δ5	αOH	*β*H	Δ^4^, *β*H, Δ^5^	αOH, αH
5(C-6)	NSD		Δ5, O, H	αOH, *β*OH		NSD	
6(C-11)	NSD		αOH, O	*β*OH, H		NSD	
7(C-12)	O		H	NSD		NSD	
8(C-17)	NSD		NSD			NSD	
9(C-25)	NSD		NSD			R	S

NSD: no statistical differences; Better DS → indicates what functional group or stereochemistry present more pronounced effects at docking score, R and S indicate the stereochemistry of C-25.

**Table 3 ijms-25-08588-t003:** ADMETx properties predicted in QikProp of the spirostans selected as inhibitors of CHK1 and CHK2.

Spirostan	CHK	CNS ^1^	Molecular Weight	Hydrogen BondD/A ^2^	QPlog Po/w ^3^	#metab ^4^	QPlogHERG ^5^	QPlogKp ^6^	Lipinski’s Rule Violations
Sp137	1, 2	0	428.611	1.00/5.20	4.893	2	−4.233	−3.155	0
Sp3	1	1	414.627	1.00/3.20	6.074	3	−4.3	−2.219	1
Sp24	1	0	430.626	2.00/4.90	5.071	4	−4.447	−2.913	1
Sp97	1	0	430.626	2.00/4.90	5.071	4	−4.447	−2.913	1
Sp1	1	1	414.627	1.00/3.20	6.123	3	−4.407	−2.212	1
Sp131	1	−1	448.642	3.00/5.65	4.574	3	−4.062	−2.937	0
Sp155	1	0	428.611	1.00/4.25	5.381	3	−4.282	−2.787	1
Sp142	1	0	428.611	1.00/5.20	4.909	2	−4.458	−3.215	0
Sp154	1	0	428.611	1.00/4.25	5.437	3	−4.417	−2.793	1
Sp101	1	0	430.626	1.00/5.20	4.835	3	−3.765	−3.034	0
Sp53	1	0	432.642	2.00/4.90	5.112	2	−4.174	−2.847	1
Sp67	1	0	432.642	2.00/4.90	5.055	2	−4.042	−2.843	1
Sp37	1	0	428.611	1.00/5.20	5.064	5	−4.146	−2.702	1
Sp13	2	0	428.611	1.00/5.20	5.125	4	−4.26	−2.692	1
Sp28	2	0	430.626	2.00/4.90	4.945	4	−4.072	−2.645	0
Sp45	2	0	432.642	2.00/4.90	5.060	2	−4.215	−2.911	1
Sp41	2	0	430.626	1.00/5.20	5.155	2	−4.144	−2.794	1
Sp14	2	0	430.626	2.00/4.90	5.142	4	−4.15	−2.59	1
Sp5	2	1	416.643	1.00/3.20	6.219	1	−4.267	−2.188	1
Sp44	2	0	430.626	1.00/5.20	5.134	3	−4.097	−2.662	1
Sp16	2	0	428.611	1.00/5.20	5.120	5	−4.144	−2.575	1
Sp65	2	0	430.626	1.00/5.20	5.095	3	−4.036	−2.813	1
Sp139	2	0	428.611	1.00/5.20	4.909	2	−4.35	−3.061	0
Sp35	2	0	430.626	2.00/4.90	5.080	4	−4.147	−2.726	1
Sp106	2	0	432.642	2.00/4.90	5.028	2	−4.164	−2.977	1

^1^ CNS—Central Nervous System activity, ^2^ Hydrogen Bond D/A—hydrogen bond donor/hydrogen bond acceptor, ^3^ QPlog Po/w—predicted octanol/water partition coefficient, ^4^ #metab—number of likely metabolic reactions, ^5^ QPlogHERG—predicted IC_50_ value for blockage of HERG K+ channels, and ^6^ QPlogKp—predicted skin permeability.

**Table 4 ijms-25-08588-t004:** HOMO-LUMO (kcal/mol) analysis for Sp137 and Sp1 in CHK1 and Sp137 and Sp13 in CHK2.

CHK1					CHK2				
aa	Sp137		Sp1		aa	Sp137		Sp13	
	HOMO_CHK1_-LUMO_Sp137_	HOMO_Sp137_-LUMO_CHK1_	HOMO_CHK1_-LUMO_Sp1_	HOMO_Sp1_-LUMO_CHK1_		HOMO_CHK2_-LUMO_Sp137_	HOMO_Sp137_-LUMO_CHK2_	HOMO_CHK2_-LUMO_Sp13_	HOMO_Sp13_-LUMO_CHK2_
Gln013	−0.1990	−0.1953	−0.2687	−0.1966	Leu226	−0.2151	0.0184	−0.2483	−0.0147
Val023	−0.1924	−0.1990	−0.2621	−0.2003	Gly227	−0.2302	−0.2104	−0.2634	−0.2138
Ala036	−0.1969	−0.1982	−0.2666	−0.1994	Val234	−0.2145	−0.2018	−0.2477	−0.2052
Glu085	0.0166	−0.3112	−0.0530	−0.3125	Lys245	−0.3067	−0.0342	−0.3399	−0.0376
Try086	−0.1825	−0.2030	−0.2522	−0.2042	Ala247	−0.2201	−0.2024	−0.2533	−0.2058
Cys087	−0.2101	−0.1946	−0.2798	−0.1958	Leu303	−0.2168	−0.2059	−0.2500	−0.2093
Ser088	−0.2186	−0.2075	−0.2883	−0.2087	Met304	−0.1864	−0.1977	−0.2197	−0.2010
Gly089	−0.2214	−0.1850	−0.2911	−0.1862	Glu305	0.0184	−0.2999	−0.0148	−0.3033
Gly090	−0.2289	−0.2085	−0.2986	−0.2098	Gly306	−0.2215	−0.1850	−0.2547	−0.1884
Asp094	0.0021	−0.3360	−0.0675	−0.3372	Gly307	−0.2304	−0.2124	−0.2636	−0.2158
Glu134	0.0053	−0.3339	−0.0644	−0.3351	Glu308	0.0184	−0.3023	−0.0147	−0.3057
Leu137	−0.1945	−0.2023	−0.2642	−0.2036	Asp311	0.0104	−0.3091	−0.0228	−0.3125
					Leu354	−0.2124	−0.1968	−0.2456	−0.2001
					Gln358	−0.2147	−0.2010	−0.2480	−0.2044

aa: Amino acid residue; HOMO_CHK1_: Highest Occupied Molecular Orbital of CHK1; LUMO_Sp137_: Lowest Unoccupied Molecular Orbital of Sp137; HOMO_Sp137_: Highest Occupied Molecular Orbital of Sp137; LUMO_CHK1_: Lowest Unoccupied Molecular Orbital of CHK1; LUMO_Sp1_: Lowest Unoccupied Molecular Orbital of Sp1; HOMO_Sp1_: Highest Occupied Molecular Orbital of Sp1; HOMO_CHK2_: Highest Occupied Molecular Orbital of CHK2; LUMO_CHK2_: Lowest Unoccupied Molecular Orbital of CHK2; LUMO_Sp13_: Lowest Unoccupied Molecular Orbital of Sp13.

## Data Availability

Data supporting reported results can be found in manuscript and Appendix A support.

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
