# Peer review of "Correlation between Molecular Docking and the Stabilizing Interaction of HOMO-LUMO: Spirostans in CHK1 and CHK2, an In Silico Cancer Approach"

_ijms, 2024, doi:10.3390/ijms25168588_

Round 1

Reviewer 1 Report

Comments and Suggestions for Authors

The article "Correlation molecular docking and of stabilizing interaction HOMO-LUMO: Spirostan sapogenins in CHK1 and CHK2, an
in silico cancer approach" is at the first view very interesting and the title is very attractive. The text is comprehensible, and the results are presented and discussed systematically and transparently. However, the details of the MD simulations are completely missing. Even so, performing 1 MD simulation of 100 ns is hardly enough to get any interesting results. The authors should perform 3 simulations of 150 ns at least to get some reproducibility.

Author Response

Comment

The article "Correlation molecular docking and of stabilizing interaction HOMO-LUMO: Spirostan sapogenins in CHK1 and CHK2, an in silico cancer approach" is at the first view very interesting and the title is very attractive. The text is comprehensible, and the results are presented and discussed systematically and transparently. However, the details of the MD simulations are completely missing. Even so, performing 1 MD simulation of 100 ns is hardly enough to get any interesting results. The authors should perform 3 simulations of 150 ns at least to get some reproducibility.

Response

The comments are appreciated; greater detail has been included in the methodology section as well as a specific section to detail the MD simulations, indicating that 3 simulations of 10 ns were carried out to present the results. Added to this, in the supplementary material  the simulation results at 150 ns for the CHK1 complex with Sp137 and with the reference were incorporated, observing a stable behavior after 100 ns. Because of this observation, MD simulations were omitted for the other complexes.

Reviewer 2 Report

Comments and Suggestions for Authors

            The manuscript by Rosales-Lopez et al. titled “Correlation molecular docking and of stabilizing interaction HOMO-LUMO: Spirostan sapogenins in CHK1 and CHK2, an in silico cancer approach” proposes the use of spirostanic saponins as possible inhibitors of enzymes (CHK1 and CHK2). Even though the study is valid and relevant, the way the authors addressed it should not be published without major alterations to the manuscript. 

            Starting with the title and abstract. The title of the manuscript does not make sense in English, I am not sure what the authors are trying to convey. The abstract is full of grammatical mistakes and it is incorrectly written. Abbreviations such as “CHK1” and “CHK2” are used throughout the abstract without being first fully defined, as the reader who may be unfamiliar with these terms will not know their unabbreviated meaning. The sentence: “CHK1 and CHK2 are enzymes that are involved in the response of DNA damage, at this time these enzymes are one of the most important due to their relationship with cancer since their inhibition has been found to have cytotoxic effects against carcinogenic cells.” requires more clarity. It is an abstract sentence, needs to be a lot clearer. Other examples in the abstract of abbreviations needing to be detailed include: “ADMETx”, “HOMO”, “LUMO”, “HOMOCHKs”, “LUMOsp”.

            The introduction does not explain very well the problem and needs to be extended. It also has very problematic sentences such as: “These two last have a marked preference towards CHK1 in contrast to VRX0466617 which provokes CHK2 inhibition; this fact difficult its use given its low selectivity as well as its low cytotoxic effect against cancer cells [12,13].” For example, what is the PDB ID of the crystal structure utilized? The authors keep referring to substitutions as “Cx” throughout the whole manuscript but there is not a figure explaining where these substitutions take place in their structures. Another very long and ineligible sentence: “To proceed to analyze the DFT results to establish a relationship of reactive stability and interaction sites and propose stabilizing interactions by HOMO-LUMO interaction from ligand to protein and vice versa, as well as determine their ADMETx properties to present the candidates that, in addition to being good specific inhibitors of CHK1 or CHK2 as well as cytotoxic, which present low toxicity in a healthy model and good pharmacology.” 

            After the introduction, it makes sense to have included the computational details of the calculations and not the results and discussion. The authors do not explain clearly their computational details. For example, I could not find the PDB ID used for the study! Regarding the molecular dynamics calculations there is barely any details of the calculations. Which parameters were used to heat up the protein? What was used for the production run? What kind of forcefields and timesteps were used? For the DFT calculations, were there geometry optimizations, frequencies calculated?

            Regarding the results and discussion section, it is very difficult to follow some of the results. For example, in Table 2, I have no idea which carbons are being referred to, the way the numbers are written, C2, C3, C4, etc, are not explained further. Maybe a figure which shows which carbons are which? Figures 4 and 5 have horrible resolution and cannot be read properly. Also, when it is mentioned “see supplementary material”, it should be pointed out where, for either a table or figure. In section 2.3, there is a sentence: “the text following an equation need not be a new paragraph. Please punctuate equations as regular text.” I am not sure what this sentence refers to. For the HOMO LUMO studies in section 2.4, did the authors optimize the geometry? Or just used the crystal structure? This needs to be explained, because without optimizing the geometries, the structures are not in a minimum and the orbitals will be quite different too. 

            Overall, since the authors ran molecular dynamic simulations, it would be a much better assessment to calculate free binding energies from a set of frames chosen properly than just rely on docking scores. Even though docking scores are useful, if the authors want this study to have relevancy for cancer applications better methods need to be considered. 

Comments on the Quality of English Language

The English language in this manuscript needs to be reviewed. There are several ineligible sentences which hinder from the overall content of the paper. Some examples are provided in the review, but not all. 

Author Response

Comment

The manuscript by Rosales-Lopez et al. titled “Correlation molecular docking and of stabilizing interaction HOMO-LUMO: Spirostan sapogenins in CHK1 and CHK2, an in silico cancer approach” proposes the use of spirostanic saponins as possible inhibitors of enzymes (CHK1 and CHK2). Even though the study is valid and relevant, the way the authors addressed it should not be published without major alterations to the manuscript.

Response

We appreciate your comments, the document was reviewed by a native of the English language; the text was restructured for  a sharp comprehension.

Comment

Starting with the title and abstract. The title of the manuscript does not make sense in English, I am not sure what the authors are trying to convey. The abstract is full of grammatical mistakes and it is incorrectly written. Abbreviations such as “CHK1” and “CHK2” are used throughout the abstract without being first fully defined, as the reader who may be unfamiliar with these terms will not know their unabbreviated meaning. The sentence: “CHK1 and CHK2 are enzymes that are involved in the response of DNA damage, at this time these enzymes are one of the most important due to their relationship with cancer since their inhibition has been found to have cytotoxic effects against carcinogenic cells.” requires more clarity. It is an abstract sentence, needs to be a lot clearer. Other examples in the abstract of abbreviations needing to be detailed include: “ADMETx”, “HOMO”, “LUMO”, “HOMOCHKs”, “LUMOsp”.

Response

The title was modified; we appreciate your comment. In the same way, meaning of acronyms and abbreviations have been added in their first appearance in the text. The introduction has been revised to present the clearest ideas

Comment

The introduction does not explain very well the problem and needs to be extended. It also has very problematic sentences such as: “These two last have a marked preference towards CHK1 in contrast to VRX0466617 which provokes CHK2 inhibition; this fact difficult its use given its low selectivity as well as its low cytotoxic effect against cancer cells [12,13].” For example, what is the PDB ID of the crystal structure utilized? The authors keep referring to substitutions as “Cx” throughout the whole manuscript but there is not a figure explaining where these substitutions take place in their structures. Another very long and ineligible sentence: “To proceed to analyze the DFT results to establish a relationship of reactive stability and interaction sites and propose stabilizing interactions by HOMO-LUMO interaction from ligand to protein and vice versa, as well as determine their ADMETx properties to present the candidates that, in addition to being good specific inhibitors of CHK1 or CHK2 as well as cytotoxic, which present low toxicity in a healthy model and good pharmacology.”

Response

The introduction has been modified to clarify the importance of the existent data. In the same sense, the description of the function for both CHK1 and CHK2, related to cancer, has been explained in a greater detail. An image regarding steroid positions has been placed for a better comprehension of the statistical analysis and discussion of ADMETx prediction and DFT analysis.

Comment

After the introduction, it makes sense to have included the computational details of the calculations and not the results and discussion. The authors do not explain clearly their computational details. For example, I could not find the PDB ID used for the study! Regarding the molecular dynamics calculations there is barely any details of the calculations. Which parameters were used to heat up the protein? What was used for the production run? What kind of forcefields and timesteps were used? For the DFT calculations, were there geometry optimizations, frequencies calculated?

Response

The PDB codes and their corresponding citation have been added, as well as details of the methodology in terms of docking, DFT calculations and molecular dynamics simulations.

Comment

Regarding the results and discussion section, it is very difficult to follow some of the results. For example, in Table 2, I have no idea which carbons are being referred to, the way the numbers are written, C2, C3, C4, etc, are not explained further. Maybe a figure which shows which carbons are which? Figures 4 and 5 have horrible resolution and cannot be read properly. Also, when it is mentioned “see supplementary material”, it should be pointed out where, for either a table or figure. In section 2.3, there is a sentence: “the text following an equation need not be a new paragraph. Please punctuate equations as regular text.” I am not sure what this sentence refers to. For the HOMO LUMO studies in section 2.4, did the authors optimize the geometry? Or just used the crystal structure? This needs to be explained, because without optimizing the geometries, the structures are not in a minimum and the orbitals will be quite different too.

Response

Figure 3 incorporates the skeleton and substituent numbering. The document has been reordered to make the mechanism section clearer and details for the optimization of geometry, in the case of designed molecules. The use of optimized crystals has been included in the references.

Comment

Overall, since the authors ran molecular dynamic simulations, it would be a much better assessment to calculate free binding energies from a set of frames chosen properly than just rely on docking scores. Even though docking scores are useful, if the authors want this study to have relevancy for cancer applications better methods need to be considered.

Response

MM-GBSA calculations for selected spirostans and references have been included to present better selection after doing scoring

Reviewer 3 Report

Comments and Suggestions for Authors

It is typical medicinal chemistry paper. It analyses the possibility that some representatives of  spirostan sapogenins may act as  effectors of checkpoint kinases 1 and 2 (CHK1 and CHK2) and thus could be considered as potentrial anticancer agents. Paper is typical - number of studies on the use of computer-aided methods for selection of natural compounds affecting chosen proteins is unusually accumulating in recent years. These papers lack physiological studies, which dimnished their importance significantly.

Anyway, paper is well scheduled and well documented and could be published. The major obstacle here is English (especially in paragraphs devoted to modeling), which has to be corrected. Difficulties with English make also that Introduction is quite chaotic and has to be ordered.

Other errors, which require corrections are as follows:

1./ line 48: should be "(which is an example of dual inhibitor)";

2./ line 85: what iAuthors call co-crystal? by definition it is “Crystalline materials composed of two or more molecules within the same crystal lattice”. What kind of co-crystal was used? Enzyme- known inhibitor complex?;

3./ line 105: something is wrong at the beginning of the sentence;

4./ lines 137-139: the sentence "Analyzing the set of structures selected for each CHK1, a trend can be noted for CHK1 candidates with 3βOH and 3oxo, without the presence of carbonyl at C12 except 1for Sp97, this can be explained due to the interactions that each one shows with the catalytic site of CHK1" is to long and completely unclear

3./ when naming inhibitor please use consequemtly one notation, for example S097 not S 097;

4./ sentence in the lines 16-165 is to long and thus not clear;

5./ line 200: statement "...selected steroids present less than one violation.." is a jargon;

6./ line 206: fragment ". Please punctuate equations as regular text. " remained form template;

7./ the sentence in lines 218-226 is to long for sure and I jave got lost trying to understand it; the same considers the text in lines 372-377;

8./ paragraph 4.1.: pdb numbers should be given to the used crystal structures of proteins

Comments on the Quality of English Language

English is a major problem of this paper.

Author Response

It is typical medicinal chemistry paper. It analyses the possibility that some representatives of spirostan sapogenins may act as effectors of checkpoint kinases 1 and 2 (CHK1 and CHK2) and thus could be considered as potentrial anticancer agents. Paper is typical - number of studies on the use of computer-aided methods for selection of natural compounds affecting chosen proteins is unusually accumulating in recent years. These papers lack physiological studies, which dimnished their importance significantly.

Anyway, paper is well scheduled and well documented and could be published. The major obstacle here is English (especially in paragraphs devoted to modeling), which has to be corrected. Difficulties with English make also that Introduction is quite chaotic and has to be ordered.

Response

The comments are appreciated. The main focus  has been highlighted and particularly the language has been reviewed in depth, by a native English speaker, especially in the modeling section including greater detail in the used methodologies. The introduction it has been rewritten to be clearer and press more information on the relevance of CHKs with respect to cancer.

Comment

Other errors, which require corrections are as follows:

1./ line 48: should be "(which is an example of dual inhibitor)";

Response

The comment has been incorporated.

Comment

2./ line 85: what iAuthors call co-crystal? by definition it is “Crystalline materials composed of two or more molecules within the same crystal lattice”. What kind of co-crystal was used? Enzyme- known inhibitor complex?;

Response

The use of the term has been used in the literature as the cystal containing the inhibitor and the protein .

Comment

3./ line 105: something is wrong at the beginning of the sentence;

Response

The entire sentence has been restructured

Comment

4./ lines 137-139: the sentence "Analyzing the set of structures selected for each CHK1, a trend can be noted for CHK1 candidates with 3βOH and 3oxo, without the presence of carbonyl at C12 except 1for Sp97, this can be explained due to the interactions that each one shows with the catalytic site of CHK1" is to long and completely unclear

Response

The paragraph has been restructured and an image has been added to stablish  the positions of the carbons.

Comment

3./ when naming inhibitor please use consequemtly one notation, for example S097 not S 097;

Response

The assessment has been homogenized throughout the document for spirostans

Comment

4./ sentence in the lines 16-165 is to long and thus not clear;

Response

The paragraph has been restructured

Comment

5./ line 200: statement "...selected steroids present less than one violation.." is a jargon;

Response

In bioinformatics, violation is a common technical word; it has been specified to denote that steroids present 1 or 0 rule breakings

Comment

6./ line 206: fragment ". Please punctuate equations as regular text. " remained form template;

Response

The equations have been inserted according to the template

Comment

7./ the sentence in lines 218-226 is to long for sure and I jave got lost trying to understand it; the same considers the text in lines 372-377;

Response

In both cases writing has been improved

Comment

8./ paragraph 4.1.: pdb numbers should be given to the used crystal structures of proteins

Response

The information has been included in the methodology as well as additional details to explain how calculations were carried out.

Round 2

Reviewer 2 Report

Comments and Suggestions for Authors

Please read the attached file for complete revisions. 

Comments on the Quality of English Language

English still needs to be revised in several parts of the manuscript. 

Author Response

Comments 1.

The manuscript by Rosales-Lopez et al. titled “Correlation between molecular docking and the
stabilizing in-2 teraction of HOMO-LUMO: Spirostans in CHK1 and CHK2, an in silico cancer
approach ” proposes the use of spirostanic saponins as possible inhibitors of enzymes (CHK1 and
CHK2). After second revision, several parts of the manuscript still need careful addressing. 

Response 1.

The entire document has been reviewed with special attention to the writing and integration of the results.

Comments 2.

There is a weird issue with spacing, that are a lot of places in the abstract and introduction with
double spacing. For example, the first sentence of the abstract, after the period there is a comma.
HOMO and LUMO do not stand for “High” or “Low”, but highest and lowest, respectively. In the
Keywords section: there is an extra comma after a semicolon in “Natural compounds”

Response 2.

All double spaces throughout the document as a result of change control were corrected, as well as proper punctuation and updating of observations made.

Comments 3.

Section 2.7 is still incomplete, the authors state that they performed 100ns calculations, however
what was the timestep? The following sentences do not make sense: The MD simulation used the
OPLS4 force field from 100 ns”, “The 100 ns MD simulations were three times”. 

Response 3.

It has been rewritten and included the information on the number of frames and the recording interval, as well as the triplicate of the study was explained in order to analyze the reproducibility of the calculation, in the same way the information on the analysis module of the results obtained was added.

Comments 4.

In section 2.8, what is reference 31 referring to? It does not make sense, because it has nothing to
do with MM-GBSA, which has its own citations. In addition, how were the binding energies
calculated? How many frames of the simulations were utilized? Were they averaged among the 3
runs mentioned?

Response 4.

The quote included is about the use of this technique in studies with steroids, in addition, 2 references have been included about the type of calculation carried out in greater detail, in addition, the definitions of each of the variables have been included, and how the values ​​are presented. in the discussion as an average accompanied by maximum and minimum values.

Comments 5.

Section 3, what does this sentence mean: “The inhibition of CHK1 and CHK2 is an excellent
alternative against the development of pathologies such as cancer”? In line 148, the authors state:
“As CHK1 is a kinase, its function is direct, and occupys the catalytic site with a comparable and
higher score; this means that it is a viable candidate as an inhibitor”. You cannot state it is a viable
candidate by docking scores. Docking scores are a very poor computational for stating it is a viable
candidate for a cancer inhibitor.

Response 5.

The writing has been restructured to focus on how the docking score calculation serves as the first selection criterion for potentially bioactive molecules, together with other studies.

Comments 6.

In Table 2, what does Better DS mean?
Figure 5 and 6 are not legible, they need to be better resolution, this was already mentioned
previously. Either make the figures bigger, or get a better resolution. They cannot be published this
way.

Response 6.

A table footer has been added to clarify the meaning of Better DS mean, as well as the figures, the original figures are attached for the publication process, since the resolution is lowered in the submission process, which have a high resolution to allow a zoom and be able to observe all the interactions described.

Comments 7.

Figure 7 – Blatant typo in “frequency”.
In Table 4, what are the units for the HOMO-LUMO gaps? 

Response 7.

The typographical error has been corrected and the units added to table 4

Comments 8.

Figure 8 – What does this sentence mean: “In Figure 8, the HOMO and LUMO interactions can
be observed in the sense of a 318 greater gap…”? 

Response 8.

The statement has been written to be clearer in the information presented.

Comments 9.

Section 3.5 – What is DeltaGbing?! Since 3 runs were performed for the MD simulations, why is
only one run shown? Are the runs averaged? I do not understand what is going on.

The average presented as a result and the maximum and minimum value added in each one have been clarified, as well as the meaning of each variable and the typographical error corrected by its corresponding symbol.

Reviewer 3 Report

Comments and Suggestions for Authors

I had reviewed this paper previously,. The present version is far better than previous one. The paper is far better scheduled and writte, In my opinion it is ready to be published in that form now.

Author Response

Comment

I had reviewed this paper previously,. The present version is far better than previous one. The paper is far better scheduled and writte, In my opinion it is ready to be published in that form now.

Response

Your previous corrections to improve the work are appreciated.

Round 3

Reviewer 2 Report

Comments and Suggestions for Authors

Some grammatical errors that should be addressed:

Line 276 - Remove "rule" or "criteria"

Line 335 - 339 - This sentence does not make sense.

Line 387 - What does this sentence mean? Needs rephrasing.

In the conclusion (line 425), remove the quotes from "molecular docking". It was not used before.  

Comments on the Quality of English Language

The quality of the text has improved. Some minor editing is still required. 

Author Response

Comment 1

Remove "rule" or "criteria"

Response 1

the word ruler was removed, to avoid redundancy

Comment 2

Line 335 - 339 - This sentence does not make sense.

Response 2

The paragraph has been written explaining the relevance and introduction to said analysis

Comment 3

Line 387 - What does this sentence mean? Needs rephrasing.

Response 3

the text has been rephrased

Comment 4

In the conclusion (line 425), remove the quotes from "molecular docking". It was not used before.

Response 4

ready

Comment 5

Minor editing of English language required

It has been reviewed by a native English speaker in detail to detect and correct errors in English.